# Air enema reduction versus hydrostatic enema reduction for intussusceptions in children: A systematic review and meta-analysis

Lan Liu[☉], Ling Zhang[☉], Yifan Fang[☉], Yingying Yang, Wen You, Jianxi Bai, Bing Zhang, Siqi Xie[ORCID]*, Yuanyuan Fang*

Department of Pediatric Surgery, Fujian Children's Hospital (Fujian Branch of Shanghai Children's Medical Center), College of Clinical Medicine for Obstetrics & Gynecology and Pediatrics, Fujian Medical University, Fuzhou, China

☉ These authors contributed equally to this work.
* mztj1016@163.com (YF); xsq59447@sina.com (SX)

**Data Availability Statement:** All relevant data are within the manuscript and its Supporting Information files.

## Abstract

### Objectives

We conducted a comprehensive meta-analysis to compare the effectiveness and safety of fluoroscopy-guided air enema reduction (FGAR) and ultrasound-guided hydrostatic enema reduction (UGHR) for the treatment of intussusception in pediatric patients.

### Methods

A systematic review and meta-analysis were conducted on retrospective studies obtained from various databases, including PUBMED, MEDLINE, Cochrane, Google Scholar, China National Knowledge Infrastructure (CNKI), WanFang, and VIP Database. The search included publications from January 1, 2003, to March 31, 2023, with the last search done on Jan 15, 2023.

### Results

We included 49 randomized controlled studies and retrospective cohort studies involving a total of 9,391 patients, with 4,841 in the UGHR and 4,550 in the FGAR. Specifically, UGHR exhibited a significantly shorter time to reduction (WMD = -4.183, 95% CI = (-5.402, -2.964), P < 0.001), a higher rate of successful reduction (RR = 1.128, 95% CI = (1.099, 1.157), P < 0.001), and a reduced length of hospital stay (WMD = -1.215, 95% CI = (-1.58, -0.85), P < 0.001). Furthermore, UGHR repositioning was associated with a diminished overall complication rate (RR = 0.296, 95% CI = (0.225, 0.389), P < 0.001) and a lowered incidence of perforation (RR = 0.405, 95% CI = (0.244, 0.670), P < 0.001).

### Conclusion

UGHR offers the benefits of being non-radioactive, achieving a shorter reduction time, demonstrating a higher success rate in repositioning in particular, resulting in a reduced length of

**Funding:** Foundation of Fujian High-level Clinical Medical Center(Siqi Xie, ETK2023016).

**Competing interests:** The authors have declared that no competing interests exist.

postoperative hospital stay, and yielding a lower overall incidence of postoperative complications, including a reduced risk of associated perforations.

## Introductions

Intussusception stands as the most prevalent etiology of intestinal obstruction in pediatric patients. A substantial majority, approximately 75–90%, exhibit no identifiable cause and are classified as idiopathic intussusception [1–4]. This condition primarily affects the small intestine, with infrequent occurrences in the large intestine [5]. Clinical presentation typically encompasses symptoms such as abdominal pain, vomiting, and hematochezia, although the classic triad of symptoms is encountered in less than 25% of cases [6,7]. Historically, fluoroscopy-guided air enema reduction (FGAR) has served as the primary therapeutic modality for intussusception. Its prominence stems from the demonstrated efficacy and safety of enema decompression established during the 1940s and 1950s. In recent years, ultrasound-guided hydrostatic enema reduction (UGHR) has gained traction as a non-invasive, radiation-free imaging technique [8–10]. The advent of UGHR in clinical practice traces its origins back to 1982 when Kim et al. [11] first reported successful reduction of ileocolonic intussusception using warm saline enema under real-time ultrasound guidance. This approach has progressively gained popularity and involves ultrasound confirmation of the intussusception's location. A predetermined initial pressure is established, followed by ultrasound-guided injection of warm saline into the intestinal tract. Successful reduction is verified when saline flows into the intestinal tract from the ileocecal region, resulting in the manifestation of characteristic signs such as the "crab claw sign" and "honeycomb sign" [12]. Although numerous studies have indicated that UGHR has advantages such as a higher success rate of resetting, greater safety, and radiation-free procedures, these merits are considered worthy of implementation in clinical practice. However, some studies also suggest that FGAR, as a traditional treatment method, remains practical in clinical settings due to its simplicity, ease of execution, and shorter learning curve. Besides, despite the burgeoning utilization of UGHR, a notable gap persists in terms of comprehensive, large-scale systematic comparisons and analyses assessing the efficacy, safety, and long-term prognostic implications of FGAR versus UGHR. We conducted a comprehensive meta-analysis comparing the efficacy and safety of air enema reduction and hydrostatic enema reduction for the treatment of childhood intussusception. Through an extensive literature search and rigorous clinical data analysis, our study aims to present a more secure and dependable therapeutic alternative for children with intussusception, thereby furnishing clinicians with compelling diagnostic and treatment evidence.

## Methods

Reporting followed the Preferred Reporting Items for Systematic Reviews and Meta-Analysis (PRISMA) guidelines [13] (S1 Checklist). We registered the study on PROSPERO, of which the registration number was CRD42023414518.

We conducted a systematic review of studies published in PUBMED, Google Scholar, MEDLINE, Cochrane, China National Knowledge Infrastructure (CNKI), Wanfang Database, VIP Database. The search has a limit on date from Jan 1, 2003 to Mar 31, 2023, with the last search done on Jan 15, 2023. No publication restrictions or study design filters were applied. We formulated the search strategy, inclusion criteria and exclusion criteria according to the PICOS principles: (1) Type of study: randomized or non-randomized controlled trial, with the

language limited to Chinese and English; (2) Participants of the study: pediatric patients (aged <18 years) who underwent enemas due to intussusception; (3) Interventions adopted: fluoroscopic air enema or ultrasound-guided saline enemas were used; (4) The main outcome indicators: time to reset, success rate of reset, recurrence rate, and occurrence of postoperative complications; (5)The search strategy for those databases was as follows: ((enema [Title/Abstract]) AND (intussusceptions [Title/Abstract])), hydrostatic enema for intussusceptions, ((enema [Title/Abstract]) AND (intussusceptions [Title/Abstract])) AND (ultrasound [Title/Abstract]), Reference lists from related articles were also scanned to broaden the search. A hand search was performed in all six databases.

Inclusion criteria were applied as follows: (1) confirmation of intussusception diagnosis; (2) subjects aged below 18 years; (3) availability of relevant outcome measures, such as patient numbers, study design, clinical symptomatology, reset success rates, complications, and recurrence; (4) provision of suitable statistical estimates or counts; and (5) comparative investigations involving both fluoroscopy-guided air enema reduction and ultrasound-guided hydrostatic enema reduction.

Exclusion criteria were applied as follows: (1) case reports involving fewer than five cases; (2) subjects exceeding 18 years of age; (3) articles categorized as reviews or meta-analyses; (4) conference abstracts; (5) articles with insufficient data; (6) cases included that did not pertain to acute intussusception or were combined with secondary intussusception; and (7) studies lacking a direct comparison between fluoroscopy-guided air enema reduction and ultrasound-guided hydrostatic enema reduction.

The following data were extracted: the first author's name, year of publication, study type, mean age, gender distribution, patient count, primary clinical symptoms, time required for reduction, reset pressure applied, reset success rate, duration of occult blood in stool, time until recovery of bowel function, length of hospital stay, recurrence rate, and complications.

The quality assessment of randomized controlled studies (RCTs) was conducted using the Cochrane Collaboration's Risk of Bias tool [14]. Only studies with low or unclear risk of overall bias were included in the meta-analysis. Non-randomized studies underwent assessment with the Newcastle-Ottawa Scale (NOS) [15]. The NOS score, ranging from 0 to 9 stars, evaluates studies across three categories: selection, comparability, and outcome/exposure. Studies with a NOS score of ≥6 stars were deemed high quality and incorporated into our analysis. The literature retrieval and data collection were to be carried out by at least two researchers. They independently read the titles and abstracts of the literature, excluding those that were not relevant to the content of this study. Subsequently, they will carefully read the full texts according to inclusion and exclusion criteria, extracting relevant information. In case of disagreements, resolution will be sought through negotiation, or a third researcher may be consulted for assistance in making a judgment.

Statistical analysis was conducted by STATA version 16.0 and RevMan version 5.2. Relative risk (RR) was applied for dichotomous variables, and weighted mean difference (WMD) was applied for continuous variables. Some study outcomes were reported as medians with ranges or mid-quartiles with ranges. According to the methods introduced by Luo et al. [16] and Wan et al. [17], those data were converted to means with deviations, thus the results for each group are presented as the mean ± standard deviation (x± s). The $I^2$ statistic was used to test the degrees of heterogeneity, the P-value of $I^2 < 0.05$ was used to indicate high heterogeneity and vice versa. The random-effects model was applied to pool the high heterogeneity results and the fixed-effects model was used for low heterogeneity (P-value of $I^2 > 0.05$; Table 2A and 2B). Begg's Test and Egger's Test were performed to assess the risk of bias (Table 3), while Begg's funnel plots were applied. $P < 0.05$ was considered to be statistically significant in the text.

## Results

We initially identified 1231 articles through our comprehensive literature search. Prior to screening, 986 records were expunged from consideration. Subsequently, after the removal of duplicate entries, an additional 119 records were excluded following a meticulous full-text review, as they failed to satisfy our predefined inclusion criteria (Fig 1). Ultimately, our

**Identification of studies via databases and registers**

**Identification**

Records identified from Databases
(n =1231)

Records removed *before screening*:
Records excluded
(n =986)

**Screening**

Records screened
(n = 263)

Duplicate records removed
(n =95)

Reports sought for retrieval
(n = 168)

Reports not retrieved
(n = 0)

Reports assessed for eligibility
(n =168)

Reports excluded
Insufficient data in the article:(n =35)
review or meta-analyses (n =24)
Included cases were not acute intussusception or combined with secondary intussusception (n=9)
There was no comparison about fluoroscopy-guided air enema reduction versus ultrasound-guided hydrostatic enema reduction (n=51)

**Included**

Studies included in review
(n =49)

**Fig 1. Flow diagram representing the selection of study.**

analysis encompassed a total of 49 studies mostly from the Asia and Europe, involving 9391 patients, with 4841 in the ultrasound-guided hydrostatic enema reduction group (UGHR) and 4550 in the fluoroscopy-guided air enema reduction group (FGAR).

## Characteristics and risk of bias of included studies

The baseline characteristics of the 49 records, including first author, publication year, study type, number of patients, male/female sex ratio, and age of operation, are presented in Table 1. The NOS scores ranged from 6 to 8 stars, reflecting the quality of the non-randomized controlled studies (case-control and cohort studies) (S1 Table), and S1A and S1B Fig presents the Cochrane Collaboration's Risk of Bias Tool for the randomized controlled studies (RCTs) that were judged to have a low risk of bias. Table 2(A) and 2(B) show the overall analyses for dichotomous and continuous variables, respectively.

## Comparations and outcomes of the meta-analysis

**Age of operation.** Forty-five studies contributed data about UGHR and FGAR, including 8501 patients (4335 in the UGHR and 4166 in the FGAR, Table 2(B)). Random-effects model was applied because of significant heterogeneity ($I^2$ = 90.00%, P < 0.001 Table 2(B)). Meta-analysis showed no significant difference between the two groups [WMD = 0.379, 95% CI = (-0.128,0.885), P = 0.143 > 0.05].

**Duration of onset.** Twenty-nine studies contributed data about UGHR and FGAR, including 3741 patients (1961 in the UGHR and 1780 in the FGAR, Table 2(B)). Random-effects model was applied because of significant heterogeneity ($I^2$ = 97.00%, P < 0.001 Table 2(B)). Meta-analysis showed no significant difference between the two groups [WMD = -0.296, 95% CI = (-1.788,1.197), P = 0.698 > 0.05].

**Clinical symptoms.** Clinical symptoms reported in the studies primarily encompassed paroxysmal crying or abdominal pain, vomiting, the presence of an abdominal mass, and the passage of bloody stools.

Paroxysmal crying or abdominal pain: Eighteen studies contributed data about UGHR and FGAR, including 2768 patients (1446/1741 in the UGHR and 1322/1616 in the FGAR, Table 2(A)). Fixed-effects model was applied because of low heterogeneity ($I^2$ = 32.30%, P = 0.098 Table 2(A)). Meta-analysis showed no significant difference between the two groups [RR = 1.031, 95% CI = (0.995,1.068), P = 0.096 > 0.05].

Vomiting: Seventeen studies contributed data about UGHR and FGAR, including 2551 patients (1335/1718 in the UGHR and 1216/1594 in the FGAR, Table 2(A)). Fixed-effects model was applied because of low heterogeneity ($I^2$ = 0.00%, P = 0.624 Table 2(A)). Meta-analysis showed no significant difference between the two groups [RR = 0.969, 95% CI = (0.928,1.011), P = 0.149 > 0.05].

Abdominal mass: Thirteen studies contributed data about UGHR and FGAR, including 1282 patients (590/820 in the UGHR and 692/935 in the FGAR, Table 2(A)). Fixed-effects model was applied because of low heterogeneity ($I^2$ = 0.00%, P = 0.594 > 0.05 Table 2(A)). Meta-analysis showed no significant difference between the two groups [RR = 1.007, 95% CI = (0.938,1.081), P = 0.852 > 0.05].

Bloody stool: Seventeen studies contributed data about UGHR and FGAR, including 1254 patients (622/1696 in the UGHR and 632/1571 in the FGAR, Table 2(A)). Random-effects model was applied because of significant heterogeneity ($I^2$ = 55.80%, P = 0.003 Table 2(A)). Meta-analysis showed no significant difference between the two groups [RR = 0.963, 95% CI = (0.855,1.085), P = 0.536 > 0.05].

**Table 1. Baseline characteristics of 49 records with 9391 patients enrolled in the meta-analysis.**

| Name | Year | Study type | Number of patients | | Gender(male/female) | | Age(m) | |
|---|---|---|---|---|---|---|---|---|
| | | | U | F | U | F | U | F |
| Wang et al [18] | 2013 | RCT | 46 | 46 | 38/8 | 40/6 | 15±5.04 | 14.16±10.2 |
| Zhang et al [19] | 2014 | RCT | 64 | 64 | 42/22 | 40/24 | 5.89±1.12 | 6.03±1.34 |
| Guo et al [20] | 2014 | R | 352 | 230 | 198/154 | 152/78 | 3–132 | 3–60 |
| Yi et al [21] | 2015 | RCT | 39 | 39 | 25/14 | 26/13 | 24.24±8.16 | 23.76±10.68 |
| Zhong et al [22] | 2015 | RCT | 44 | 40 | '32/12 | 27/13 | 14.4±1.32 | 10.8±1.44 |
| Wu et al [23] | 2015 | R | 45 | 42 | 30/15 | 28/14 | 9.5±3.9 | 9.3±3.5 |
| Li et al [24] | 2015 | R | 76 | 73 | 51/25 | 49/24 | 14.4±6 | 13.2±7.2 |
| Jiang et al [25] | 2016 | RCT | 74 | 74 | 40/34 | 39/35 | 33.6±18 | 34.8±15.6 |
| Liao et al [26] | 2016 | RCT | 30 | 29 | 14/16 | 18/11 | 12.6±4.92 | 12.72±2.28 |
| Yang et al [27] | 2016 | RCT | 50 | 50 | 36/14 | 35/15 | 9.6±2.4 | 9.9±2.4 |
| Deng et al [28] | 2016 | RCT | 45 | 45 | 28/17 | 30/15 | 8.8±3.6 | 8.9±3.8 |
| He et al [29] | 2017 | RCT | 60 | 60 | 32/28 | 31/29 | 36±18 | 36±14.4 |
| Xu et al [30] | 2017 | R | 126 | 120 | 67/53 | 65/55 | 31.2±16.8 | 32.4±16.8 |
| Zhang et al [31] | 2017 | RCT | 34 | 34 | N | N | N | N |
| Xie et al [32] | 2017 | RCT | 62 | 62 | 40/22 | 42/20 | 23.52±6.29 | 20.67±4.14 |
| Wang et al [33] | 2018 | R | 406 | 417 | 298/108 | 305/112 | 9.5±1.7 | 11.3±4.5 |
| Yu et al [34] | 2018 | R | 45 | 45 | 22/23 | 23/22 | 30.72±7.32 | 30.72±6.48 |
| Wu et al [35] | 2018 | RCT | 62 | 62 | N | N | N | N |
| Pan et al [36] | 2018 | R | 373 | 262 | 223/150 | 168/94 | 13.1±7.3 | 12.6±6.7 |
| Deng et al [37] | 2018 | RCT | 80 | 80 | 61/19 | 55/25 | 10.15±4.75 | 9.93±4.75 |
| Zhang et al [38] | 2018 | R | 45 | 46 | 23/22 | 25/21 | 3.54±1.44 | 3.59±1.48 |
| Zhou et al [39] | 2019 | RCT | 41 | 41 | 23/18 | 25/16 | 10.11±4.15 | 10.77±4.85 |
| Zhao et al [40] | 2019 | RCT | 37 | 37 | 20/17 | 21/16 | 10.5±4.8 | 10.2±5.0 |
| Wang et al [41] | 2019 | RCT | 30 | 30 | 21/9 | 18/12 | 26.9±19.7 | 24.8±13.7 |
| Jiang et al [42] | 2019 | R | 58 | 58 | N | N | N | N |
| Wang et al [43] | 2019 | RCT | 50 | 50 | 28/22 | 27/23 | 21.48±7.56 | 17.4±9.96 |
| Zhang et al [44] | 2020 | R | 50 | 48 | 37/13 | 24/14 | 14.15±6.55 | 14.57±7.09 |
| Guo et al [45] | 2020 | R | 38 | 38 | 20/18 | 17/21 | 20.4±13.44 | 19.8±12.36 |
| Wang et al [46] | 2020 | R | 240 | 192 | N | N | 24.00±9.71 | 20.16±4.10 |
| Li et al [47] | 2020 | RCT | 45 | 45 | 28/17 | 26/19 | 29.73±7.91 | 31.24±8.59 |
| Qi et al [48] | 2020 | RCT | 35 | 35 | 20/15 | 21/14 | 19.08±3.12 | 18.6±2.76 |
| Sui et al [49] | 2021 | R | 105 | 104 | 77/28 | 68/36 | 87±13.08 | 83.64±15.84 |
| Cai et al [50] | 2021 | RCT | 23 | 22 | 12/11 | 12/10 | 1.62±0.45 | 1.59±0.45 |
| Ding et al [51] | 2021 | RCT | 31 | 31 | 21/10 | 20/11 | 15.66±2.73 | 19.45±2.37 |
| Zhang et al [52] | 2021 | RCT | 76 | 72 | 45/31 | 49/23 | 42.24±7.32 | 40.80±6.48 |
| Chen et al [53] | 2021 | R | 42 | 42 | 23/19 | 22/20 | 11.76±5.04 | 11.4±4.92 |
| Lian et al [54] | 2021 | RCT | 49 | 49 | 27/22 | 29/20 | 20.16±6.6 | 19.92±7.56 |
| Chen et al [55] | 2021 | RCT | 40 | 40 | 23/17 | 24/16 | 12.36±3.96 | 12.24±3.96 |
| Du et al [56] | 2021 | R | 45 | 42 | 29/16 | 27/15 | 13.65±4.27 | 14.78±5.02 |
| Pei et al [57] | 2021 | R | 43 | 43 | 25/18 | 24/19 | 22.33±4.55 | 21.09±4.38 |
| Liu et al [58] | 2021 | P | 1119 | 1005 | 731/388 | 670/335 | 24.38±23.78 | 25.80±21.99 |
| Yang et al [12] | 2021 | R | 119 | 245 | 89/30 | 163/82 | 25.13±2.03 | 22.47±1.52 |
| Han et al [59] | 2022 | RCT | 90 | 90 | 68/22 | 54/36 | 8.3±1.6 | 8.5±1.7 |
| Liu et al [60] | 2022 | RCT | 35 | 35 | 20/15 | 19/16 | 37.01±3.24 | 36.01±3.31 |
| Lv et al [61] | 2022 | R | 43 | 37 | 30/13 | 23/14 | 12.01±1.20 | 11.82±0.92 |
| Liu et al [62] | 2022 | RCT | 58 | 58 | 31/27 | 30/28 | 16.23±1.85 | 15.26±2.05 |

*(Continued)*

**Table 1.** (Continued)

| Name | Year | Study type | Number of patients | | Gender(male/female) | | Age(m) | |
|---|---|---|---|---|---|---|---|---|
| | | | U | F | U | F | U | F |
| Pu et al [63] | 2022 | RCT | 75 | 75 | 46/29 | 45/30 | 12.32±3.15 | 12.23±3.12 |
| Chukwu et al [64] | 2022 | RCT | 26 | 26 | 16/10 | 19/7 | 5.5±1.8 | 6.1±1.6 |
| Lian et al [65] | 2023 | RCT | 40 | 40 | 29/11 | 27/13 | 13.68±10.01 | 13.03±7.33 |

R, retrospective cohort study; RCT, randomized controlled trial study; P, prospective cohort study; N: Not reported; m: Month; U, ultrasound-guided hydrostatic enema reduction; F, fluoroscopy-guided air enema reduction.

**Outcomes.** The primary outcome measures for enema reduction in cases of intussusception comprise resetting time, resetting pressure, success rate of reduction, duration of occult blood in stool, length of hospitalization, and recurrence.

Resetting time: Thirty-one studies contributed data about UGHR and FGAR, including 4236 patients (2146 in the UGHR and 2090 in the FGAR, Table 2(B)). Random-effects model

**Table 2. Pooled proportions of clinical characteristics for dichotomous variables (A). Pooled proportions of clinical characteristics for continuous variables (B).**

| Outcome | Number of studies | Participates (n) | | Total number of cases (N) | | Statistical results | | | Heterogeneity | | Analysis model |
|---|---|---|---|---|---|---|---|---|---|---|---|
| | | U | F | U | F | Statistic | Value(95%CI) | P value | $I^2$ (%) | P value | |
| Male | 45 | 2839 | 2712 | 4447 | 4204 | RR | 0.994(0.964,1.026) | 0.718 | 0.00 | 0.988 | Fixed |
| Female | 45 | 1608 | 1492 | 4447 | 4204 | RR | 1.010(0.955,1.069) | 0.720 | 0.00 | 0.993 | Fixed |
| Paroxysmal crying or Abdominal pain | 18 | 1446 | 1322 | 1741 | 1616 | RR | 1.031(0.995,1.068) | 0.096 | 32.30 | 0.098 | Fixed |
| Vomiting | 17 | 1335 | 1216 | 1718 | 1594 | RR | 0.969(0.928,1.011) | 0.149 | 0.00 | 0.624 | Fixed |
| Abdominal mass | 13 | 590 | 692 | 820 | 935 | RR | 1.007(0.938,1.081) | 0.852 | 0.00 | 0.594 | Fixed |
| Bloody stool | 17 | 622 | 632 | 1696 | 1571 | RR | 0.963(0.855,1.085) | 0.536 | 55.80 | 0.003 | Random |
| Success rate of reset | 48 | 4518 | 3766 | 4722 | 4305 | RR | 1.128(1.099,1.157) | <0.001※ | 71.40 | <0.001※ | Random |
| Recurrence | 25 | 186 | 293 | 3134 | 2680 | RR | 0.391(0.269,0.569) | <0.001※ | 51.50 | 0.002 | Random |
| Total complications | 20 | 58 | 195 | 1349 | 1225 | RR | 0.296(0.225,0.389) | <0.001※ | 13.30 | 0.288 | Fixed |
| Perforation | 23 | 13 | 43 | 2376 | 2381 | RR | 0.405(0.244,0.670) | <0.001※ | 0.00 | 0.968 | Fixed |
| Vomiting | 10 | 14 | 32 | 619 | 563 | RR | 0.463(0.271,0.791) | 0.050 | 0.00 | 0.825 | Fixed |
| Diarrhea | 9 | 13 | 43 | 558 | 507 | RR | 0.318(0.182,0.558) | <0.001※ | 0.00 | 0.948 | Fixed |

| Outcome | Number of studies | Total number of cases (N) | | Statistical results | | | Heterogeneity | | Analysis model |
|---|---|---|---|---|---|---|---|---|---|
| | | U | F | Statistic | Value(95%CI) | P value | $I^2$ (%) | P value | |
| Age | 45 | 4335 | 4166 | WMD | 0.379(-0.128,0.885) | 0.143 | 90.00 | <0.001※ | Random |
| Duration of onset | 29 | 1961 | 1780 | WMD | -0.296(-1.788,1.197) | 0.698 | 97.00 | <0.001※ | Random |
| Resetting time | 31 | 2146 | 2090 | WMD | -4.183(-5.402,-2.964) | <0.001※ | 98.60 | <0.001※ | Random |
| Resetting pressure | 4 | 234 | 360 | WMD | 1.550(-0.292,3.392) | 0.099 | 99.80 | <0.001※ | Random |
| Duration of occult blood in stool | 7 | 435 | 431 | WMD | -0.808(-1.098,-0.517) | <0.001※ | 89.70 | <0.001※ | Random |
| Length of hospitalization | 18 | 1772 | 1780 | WMD | -1.215(-1.58,-0.85) | <0.001※ | 99.40 | <0.001※ | Random |

RR, relative risk; CI, confidence interval; U, ultrasound-guided hydrostatic enema reduction; F, fluoroscopy-guided air enema reduction; ※, P < 0.05 was considered to be statistically significant.

WMD, weighted mean difference; CI, confidence interval; U, ultrasound-guided hydrostatic enema reduction; F, fluoroscopy-guided air enema reduction; ※, P < 0.05 was considered to be statistically significant.

was applied because of significant heterogeneity ($I^2$ = 98.60%, P < 0.001 Table 2(B)). Meta-analysis showed significant difference between the two groups [WMD = -4.183, 95% CI = (-5.402, -2.964), P < 0.001; S2 Fig], which demonstrated significantly less resetting time of UGHR.

Resetting pressure: Four studies contributed data about UGHR and FGAR, including 594 patients (234 in the UGHR and 360 in the FGAR, Table 2(B)). Random-effects model was applied because of significant heterogeneity ($I^2$ = 99.80%, P < 0.001 Table 2(B)). Meta-analysis showed significant difference between the two groups [WMD = 1.55, 95% CI = (-0.292,3.392), P = 0.099 > 0.05], which demonstrated significantly less resetting time of UGHR.

Success rate of reset: Forty-eight studies contributed data about UGHR and FGAR, including 8284 patients (4518/4722 in the UGHR and 3766/4305 in the FGAR, Table 2(A)). Random-effects model was applied because of significant heterogeneity ($I^2$ = 71.40%, P < 0.001 Table 2(A)). Meta-analysis showed significant difference between the two groups [RR = 1.128, 95% CI = (1.099,1.157), P < 0.001; S3 Fig], which demonstrated significantly higher reset success rate of UGHR.

Duration of occult blood in stool: Seven studies contributed data about UGHR and FGAR, including 866 patients (435 in the UGHR and 431 in the FGAR, Table 2(B)). Random-effects model was applied because of significant heterogeneity ($I^2$ = 89.70%, P < 0.001 Table 2(B)). Meta-analysis showed significant difference between the two groups [WMD = -0.808, 95% CI = (-1.098, -0.517), P < 0.001], which demonstrated significantly shorter duration of occult blood in stool of UGHR.

Length of hospitalization: Eighteen studies contributed data about UGHR and FGAR, including 3552 patients (1772 in the UGHR and 1780 in the FGAR, Table 2(B)). Random-effects model was applied because of significant heterogeneity ($I^2$ = 99.40%, P < 0.001 Table 2(B)). Meta-analysis showed significant difference between the two groups [WMD = -1.215, 95% CI = (-1.58, -0.85), P < 0.001; S4 Fig], which demonstrated significantly shorter length of hospitalization of UGHR.

Recurrent rate: Twenty-five studies contributed data about UGHR and FGAR, including 479 patients (186/3134 in the UGHR and 293/2680 in the FGAR, Table 2(A)). Random-effects model was applied because of significant heterogeneity ($I^2$ = 51.50%, P < 0.001 Table 2(A)). Meta-analysis showed significant difference between the two groups [RR = 0.391, 95% CI = (0.269,0.569), P = 0.002<0.05; S5 Fig], which demonstrated significantly less relapse rate of UGHR.

## Complications

To describe the occurrence of complications during the enema reduction procedure for intussusception, we calculated the overall complication rate, perforation rate, as well as rates of vomiting and diarrhea.

Total complications rate: Twenty studies contributed data about UGHR and FGAR, including 253 patients (58/1349 in the UGHR and 195/1225 in the FGAR, Table 2(A)). Fixed-effects model was applied because of low heterogeneity ($I^2$ = 13.30%, P = 0.288 Table 2(A)). Meta-analysis showed significant difference between the two groups [RR = 0.296, 95% CI = (0.225,0.389), P < 0.001; S6 Fig], which demonstrated significantly lower total complications rate of UGHR.

Perforation rate: Twenty-three studies contributed data about UGHR and FGAR, including 56 patients (13/ 2376 in the UGHR and 43/2381 in the FGAR, Table 2(A)). Fixed-effects model was applied because of low heterogeneity ($I^2$ = 0.00%, P = 0.968 Table 2(A)). Meta-analysis showed significant difference between the two groups [RR = 0.405, 95% CI = (0.244,0.670), P < 0.001; S7 Fig], which demonstrated significantly lower perforation rate of UGHR.

Incidence of post-operative vomiting: Ten studies contributed data about UGHR and FGAR, including 46 patients (14/619 in the UGHR and 32/563 in the FGAR, Table 2(A)). Fixed-effects model was applied because of low heterogeneity ($I^2 = 0.00\%$, P = 0.825 Table 2 (A)). Meta-analysis showed significant difference between the two groups [RR = 0.463 , 95% CI = (0.271,0.791), P < 0.001], which demonstrated significantly lower post-operative vomiting rate of UGHR.

Incidence of post-operative diarrhea: Nine studies contributed data about UGHR and FGAR, including 56 patients (13/ 558 in the UGHR and 43/507 in the FGAR, Table 2(A)). Fixed-effects model was applied because of low heterogeneity ($I^2 = 0.00\%$, P = 0.948 Table 2 (A)). Meta-analysis showed significant difference between the two groups [RR = 0.318 , 95% CI = (0.182,0.558), P < 0.001], which demonstrated significantly lower post-operative diarrhea rate of UGHR.

## Publication bias

Begg's Test and Egger's Test were performed, and Begg's funnel plots were generated for some of the included records. Different subgroups were defined to assess publication bias (Table 3). Several largely symmetrical inverted funnel plots were observed (S8A–S8D Fig), and publications displaying significant bias were removed.

**Table 3. Begg's and Egger's test of publication bias of clinical characteristics.**

| Outcome | Number of studies | P-value[a] | |
|---|---|---|---|
| | | Begg'test | Egger'test |
| **Gender** | | | |
| Male | 45 | 0.883 | 0.388 |
| Female | 45 | 0.604 | 0.117 |
| Age | 45 | 0.087 | 0.893 |
| Duration of onset | 29 | 0.003* | 0.485 |
| **Clinical symptoms** | | | |
| Paroxysmal crying or abdominal pain | 18 | 0.127 | 0.025 |
| Vomiting | 17 | 0.753 | 0.462 |
| Abdominal mass | 13 | 1.000 | 0.607 |
| Bloody stool | 17 | 0.174 | 0.249 |
| **Ending indicators** | | | |
| Resetting time | 31 | 0.248 | 0.004* |
| Resetting pressure | 4 | 0.734 | 0.378 |
| Success rate of reset | 48 | 0.001* | 0.000* |
| Duration of occult blood in stool | 7 | 0.764 | 0.811 |
| Length of hospitalization | 18 | 0.069 | 0.676 |
| Recurrence | 25 | 0.216 | 0.618 |
| **Complications** | | | |
| Total complications | 20 | 0.456 | 0.845 |
| Perforation | 23 | 0.128 | 0.236 |
| Vomiting | 10 | 0.371 | 0.795 |
| Diarrhea | 9 | 0.348 | 0.166 |

[a]: P value means the value of Pr>|z| (continuity corrected, in Begg's Test) or P>|t| (in Egger's Test)

*:P value < 0.05 was considered to have a high risk of publication bias.

## Discussions

Pediatric intussusception is characterized by the invagination of one segment of the bowel into an immediately adjacent segment, resulting in the obstruction of intestinal contents. Over time, compromised vascular flow to the affected segment can lead to ischemia, necrosis, and potentially perforation [10,66]. Therefore, early diagnosis and prompt treatment are imperative to improve prognosis. While radiological imaging plays a pivotal role in diagnosing and treating this condition, it is often not the initial choice in clinical practice due to concerns regarding radiation exposure. Ultrasound, conversely, stands out as the preferred imaging modality for diagnosis owing to its remarkable specificity (88%-100%), high sensitivity (98%-100%), and absence of ionizing radiation [67–69]. In cases of uncomplicated pediatric intussusception, imaging-guided enema reduction stands as the globally recognized standard for nonsurgical treatment [70]. To evaluate the efficacy and safety of ultrasound-guided hydrostatic enema reduction (UGHR) versus fluoroscopy-guided air enema reduction (FGAR), we conducted a comprehensive analysis encompassing clinical presentations, outcome parameters, and postoperative complications in both groups. Our primary objective is to equip healthcare practitioners with valuable insights for making informed treatment decisions when managing patients with intussusception.

We enrolled a total of 49 studies into our analysis, of which was based on a mixture of randomized and non-randomized trials. The outcomes of the meta-analysis concerning clinical presentations of intussusception, including paroxysmal crying and abdominal pain, the presence of an abdominal mass, time of onset and the occurrence of blood in stools, consistently indicated no significant differences when comparing the two groups.

The findings of this meta-analysis indicate that UGHR is characterized by a shorter resetting time, a higher success rate of reset, and a reduced duration of hospitalization (Table 2A and 2B). It has been proposed that during the UGHR procedure, real-time ultrasound enables the observation of the gradual movement of the intussusception towards the ileocecal region. During this phase, increasing the enema pressure can enhance the repositioning success rate and decrease the repositioning time. Additionally, the use of warm saline aids in the expulsion of intestinal contents, reducing the absorption of toxins by the intestinal tract. This, in turn, mitigates complications in children following the enema reduction, ultimately leading to a shorter hospital stay [12,36,41,62,71].

Complications arising from intussusception enema reduction are a critical aspect of assessing its safety, with intestinal perforation being one of the most severe complications [72]. During air enema, when the intestinal lumen pressure is high, the intestinal tube undergoes significant expansion. If excessive or sudden pressure is applied, air entering the terminal ileum may result in a tense pneumoperitoneum, potentially leading to intestinal perforation [73]. It has been reported [74] that UGHR may be less hygienic and could lead to intra-abdominal fecal contamination in case of intestinal perforation, which, if not promptly treated, can result in severe complications and endanger the patient's life. The meta-analysis presented in this article demonstrates that UGHR repositioning is associated with a lower overall complication rate, including a lower incidence of perforation (Table 2A). Furthermore, the occurrence of postoperative vomiting and diarrhea is significantly reduced in children. Pan et al [36] suggest that this reduction in complications may be attributed to the slower movement of the water column during the water enema, causing less damage to the intestinal mucosa and possessing some mucosal dialysis function, resulting in a lower incidence of postoperative complications. Additionally, UGHR enables the measurement of intestinal tube hemodynamics, observation of the intestinal wall's blood supply, and determination of its viability. This can effectively mitigate the risk of perforation due to high pressure during the enema procedure

[41,63]. It is recommended to employ intermittent ultrasound monitoring to assess the intestinal canal diameter during enema operations, reducing the likelihood of perforation. UGHR also allows for clear visualization of the intussusception mass and early detection of pathological predisposing points or residual intussusception. Overall, it can be inferred that UGHR provides significant advantages in the treatment of intussusception in children.

However, the main disadvantage of UGHR is that the success of its enemas is significantly related to the experience of the operator, which requires pediatric surgeons to be taught and trained in ultrasound or radiology. This study exhibits several limitations too. Firstly, it's worth noting that most studies included in this analysis were single-center trials. While our overall sample size is substantial, single-center studies may induce inevitable biases. Secondly, it's noteworthy that the surgical team was also involved in authoring the reports. This potential author-surgeon bias should be taken into consideration when interpreting the findings. Thirdly, certain outcome measures, such as repositioning pressure, duration of postoperative blood in the stool, and postoperative vomiting or diarrhea, exhibited lower reliability due to a limited number of reported studies, resulting in a relatively small sample size for these specific parameters. Lastly, the enrolled studies were mostly from the Asia and Europe, an inevitable selection bias was existed.

## Conclusions

In conclusion, it can be affirmed that both UGHR and FGAR represent safe and effective non-surgical approaches for the management of pediatric acute intussusception. However, when comparing the two methods, UGHR emerges as the preferable choice. This preference is rooted in its nonradioactive nature, quicker repositioning times, higher success rates in repositioning, reduced postoperative hospitalization durations, fewer overall postoperative complications, and a notably decreased incidence of concurrent perforation when compared to FGAR.

## Supporting information

**S1 Checklist. PRISMA 2020 checklist.**
(DOCX)

**S1 Fig.** A. Risk of bias summary graph 1 for the included randomized controlled trial. B. Risk of bias summary graph 2 for the included randomized controlled trial.
(TIF)

**S2 Fig. Comparisons of resetting time in ultrasound-guided hydrostatic enema reduction (UGHR) and fluoroscopy-guided air enema reduction (FGAR).**
(TIF)

**S3 Fig. Comparisons of the rate of successful reset in ultrasound-guided hydrostatic enema reduction (UGHR) and fluoroscopy-guided air enema reduction (FGAR).**
(TIF)

**S4 Fig. Comparisons of the length of hospitalization in ultrasound-guided hydrostatic enema reduction (UGHR) and fluoroscopy-guided air enema reduction (FGAR).**
(TIF)

**S5 Fig. Comparisons of the rate of recurrent in ultrasound-guided hydrostatic enema reduction (UGHR) and fluoroscopy-guided air enema reduction (FGAR).**
(TIF)

**S6 Fig. Comparisons of the rate of total complications in ultrasound-guided hydrostatic enema reduction (UGHR) and fluoroscopy-guided air enema reduction (FGAR).** (TIF)

**S7 Fig. Comparisons of the rate of perforation in ultrasound-guided hydrostatic enema reduction (UGHR) and fluoroscopy-guided air enema reduction (FGAR).** (TIF)

**S8 Fig.** A. Meta-analysis of male between UGHR and FGAR. B. Meta-analysis of the rate of perforation between UGHR and FGAR. C. Meta-analysis of vomiting between UGHR and FGAR. D. Meta-analysis of age between UGHR and FGAR. (TIF)

**S1 Table. Newcastle-Ottawa scale scores for non-randomized controlled studies.** (DOCX)

**S1 Dataset. Minimal dataset underlying the results.** (XLSX)

## Author Contributions

**Conceptualization:** Yingying Yang.

**Data curation:** Lan Liu, Yingying Yang.

**Funding acquisition:** Yifan Fang, Bing Zhang.

**Investigation:** Wen You.

**Methodology:** Wen You.

**Project administration:** Ling Zhang, Wen You.

**Resources:** Ling Zhang, Yifan Fang.

**Software:** Lan Liu.

**Supervision:** Jianxi Bai.

**Validation:** Ling Zhang.

**Visualization:** Yifan Fang, Yuanyuan Fang.

**Writing – original draft:** Lan Liu, Siqi Xie, Yuanyuan Fang.

**Writing – review & editing:** Siqi Xie, Yuanyuan Fang.

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
