## [Decision Letter · Decision Letter 0]

8 Dec 2023

PONE-D-23-36429Air enema reduction versus hydrostatic enema reduction for intussusceptions in children: An updated systematic review and meta-analysisPLOS ONE

Dear Dr. Xie,

Thank you for submitting your manuscript to PLOS ONE. After careful consideration, we feel that it has merit but does not fully meet PLOS ONE’s publication criteria as it currently stands. Therefore, we invite you to submit a revised version of the manuscript that addresses the points raised during the review process.

**ACADEMIC EDITOR: **

This systematic review and meta-analysis presents the systematic review results of the management of pediatric intussusception with ultrasound-guided hydrostatic reduction vs fluoroscopic-guided air reduction. The study protocol was registered and follows PRISMA guidelines. Overall it is a well-written manuscript and that comparison of two management approach is worthwhile. However, the manuscript need revision regarding following besides the ones addressed by the reviewers:

The methodology needs to be revised. The details of search strategy, definition of PICOS, the inclusion and exclusion criteria should be addressed more clearly for the readers. In addition, the selection process and data extraction process should be given clearly.Regarding screening process, the reason for exclusion of the studies should be given clearly in order to prevent bias while interpreting the results.When interpreting the results, it will be better to comment on the distribution of the countries in which the included studies were published.Regarding the funding, the authors stated that there is no funding during submission, they stated that there is funding in the end of the article. Please verify clearly.   ==============================

We look forward to receiving your revised manuscript.

Kind regards,

Ozlem Boybeyi-Turer

Academic Editor

PLOS ONE

5. We are unable to open your Supporting Information file [supplementary Table S1.docx]. Please kindly revise as necessary and re-upload.

Additional Editor Comments:

This systematic review and meta-analysis presents the systematic review results of the management of pediatric intussusception with ultrasound-guided hydrostatic reduction vs fluoroscopic-guided air reduction. The study protocol was registered and follows PRISMA guidelines. Overall it is a well-written manuscript and that comparison of two management approach is worthwhile. However, the manuscript need revision regarding following besides the ones addressed by the reviewers:

• The methodology needs to be revised. The details of search strategy, definition of PICOS, the inclusion and exclusion criteria should be addressed more clearly for the readers. In addition, the selection process and data extraction process should be given clearly.

• Regarding screening process, the reason for exclusion of the studies should be given clearly in order to prevent bias while interpreting the results.

• When interpreting the results, it will be better to comment on the distribution of the countries in which the included studies were published.

• Regarding the funding, the authors stated that there is no funding during submission, they stated that there is funding in the end of the article. Please verify clearly.

Reviewers' comments:

Reviewer's Responses to Questions

**Comments to the Author**

1. Is the manuscript technically sound, and do the data support the conclusions?

Reviewer #1: Yes

Reviewer #2: Partly

2. Has the statistical analysis been performed appropriately and rigorously? 

Reviewer #1: Yes

Reviewer #2: Yes

3. Have the authors made all data underlying the findings in their manuscript fully available?

Reviewer #1: Yes

Reviewer #2: Yes

4. Is the manuscript presented in an intelligible fashion and written in standard English?

Reviewer #1: Yes

Reviewer #2: Yes

5. Review Comments to the Author

Reviewer #1: Reviewer Comments

RE: PONE-D-23-36429

The authors of this manuscript gave a detailed systematic review and meta-analysis of studies done over a period of 20 years (2003 – 2023) on non-operative management of intussusception. The objective was to compare the efficacy and outcome of fluoroscopy-guided air enema

reduction (FGAR) and ultrasound-guided hydrostatic enema reduction (UGHR) modalities in the treatment of intussusception in children. The manuscript is well written, and the language used is easy to comprehend. To improve on the quality of the manuscript, I suggest the following revisions.

Title: The title is better as “Air enema reduction versus hydrostatic enema reduction for intussusceptions in children: A systematic review and meta-analysis”. Avoid such terms as updated or comprehensive. Meta-analysis is expected to be comprehensive and to provide updated information

Introduction: There is need for more details on rationale for the study, a study question, and a hypothesis

Methods: Was any article excluded based on the language of publication? How many authors independently reviewed the article from the initial search, and critical review of included articles? How did the authors address inter-rater variability in the review of included articles?

Discussion: Repeat of information already in the methods and results sections. I would like a succinct summary of your main findings.

References: The referencing style is not uniform

Reviewer #2: This systematic review and meta-analysis address the management of pediatric intussusception with ultrasound-guided hydrostatic reduction (UGHR) vs fluoroscopic-guided air reduction (FGAR). The trial was registered and follows PRISMA guidelines. Authors assessed included articles for bias and focused on high quality studies although the bulk of included studies are non-randomized cohorts. The main claim of the paper is the superiority of UGHR. Given the frequency of intussusception in pediatrics, it is helpful to compare the two methods of reduction although most institutions have limitations in training and/or availability of reduction 24/7 that dictates the institutional preferred method. The main concern with regards to the methodology of this paper is the initial screening and selection of articles. Authors note 1231 articles were identified through a comprehensive literature search but “prior to screening, 986 records expunged from consideration”. Unless it is made clear how and why these papers were expunged prior to screening, this review and meta-analysis is incomplete and results may be inaccurate due to an exclusion of the bulk of the literature identified without screening for relevance so I would invite the authors to screen all relevant literature in the next revision.

The authors review the two methods of reduction in the introduction and discussion. As noted above, the included literature in the search seems to be a sub-selection without appropriate screening of all literature identified by the initial search. I would request clarification regarding the 986 articles that were identified but expunged prior to screening. I would suggest the authors screen these 986 articles using their inclusion and exclusion criteria to ensure all relevant literature is considered.

It is helpful that the authors have assessed the risk of bias for individual studies. I would consider highlighting in the discussion that the literature is based on a mix of randomized and non-randomized trials. Specifically, the fact that one very large retrospective cohort study contributes 1119 patients to the total of 9391 (and most of the included randomized trials are much smaller than the non-randomized). Although this information is available in Table 1, I would suggest making it more accessible to readers.

Overall this is a well-written manuscript and the comparison of UGHR and FGAR is worthwhile; however, revision is required to include screening of all appropriate literature as there may be selection bias with the current methods that do not explain why only a subset of literature underwent screening.

6. PLOS authors have the option to publish the peer review history of their article (what does this mean?). If published, this will include your full peer review and any attached files.

Reviewer #1: No

Reviewer #2: No

---

## [Author Response · Author response to Decision Letter 0]

2 Jan 2024

# ACADEMIC EDITOR

Q 1: The methodology needs to be revised. The details of search strategy, definition of PICOS, the inclusion and exclusion criteria should be addressed more clearly for the readers. In addition, the selection process and data extraction process should be given clearly.

A 1: Thank you very much for your valuable suggestions, I will answer your questions in detail gladly.

Detailed part of the search strategy: we formulated the search strategy, inclusion criteria and exclusion criteria according to the PICOS principles: (1) Type of study: randomized or non-randomized controlled trial, with the language limited to Chinese and English; (2) Participants of the study: pediatric patients (aged <18 years) who underwent enemas due to intussusception; (3) Interventions adopted: fluoroscopic air enema or ultrasound-guided saline enemas were used; (4) The main outcome indicators: time to reset, success rate of reset, recurrence rate, and occurrence of postoperative complications. (5) Search strategy: Take Pubmed as an example: a specific search strategy is shown (see the suppl-figure 1 and revised-Figure 1 in the manuscript), and we have revised the text in red (page 5, line 2-9).

Literature search and data collection part: The literature retrieval and data collection are to be carried out by at least two researchers. They will independently read the titles and abstracts of the literature, excluding those that are not relevant to the content of this study. Subsequently, they will carefully read the full texts according to inclusion and exclusion criteria, extracting relevant information. In case of disagreements, resolution will be sought through negotiation, or a third researcher may be consulted for assistance in making a judgment. (page 6, line 21-22, page 7, line 1-6)

Q 2: Regarding screening process, the reason for exclusion of the studies should be given clearly in order to prevent bias while interpreting the results.

A 2: I am so grateful for the valuable advice, the literature screening process involved 2 researchers independently reading the titles and abstracts of the literature to initially exclude literature that did not match the content of this study for the specific reasons for exclusion shown in revised-Figure 1 in the manuscript

Q 3: When interpreting the results, it will be better to comment on the distribution of the countries in which the included studies were published.

A 3: Thank you for this valuable suggestion. We have added associated information in the result and the limitations.

Q 4: Regarding the funding, the authors stated that there is no funding during submission, they stated that there is funding in the end of the article. Please verify clearly. 

A 4: I am very sorry for your misunderstanding due to my negligence during submission, this article does have financial support and we will make changes subsequently.

# Reviewer 1

Q 1: Title: The title is better as “Air enema reduction versus hydrostatic enema reduction for intussusceptions in children: A systematic review and meta-analysis”. Avoid such terms as updated or comprehensive. Meta-analysis is expected to be comprehensive and to provide updated information

A1: Thank you very much for your valuable suggestions, the title of this article has been changed. (see the title)

Q 2: Introduction: There is need for more details on rationale for the study, a study question, and a hypothesis.

A2: Thank you very much for your valuable suggestions, we have revised the article's introduction. (see the introduction in red）

Q 3: Methods: Was any article excluded based on the language of publication? How many authors independently reviewed the article from the initial search, and critical review of included articles? How did the authors address inter-rater variability in the review of included articles?

A 3: I am so grateful for the valuable advice. We formulated the search strategy, inclusion criteria and exclusion criteria according to the PICOS principles: (1) Type of study: randomized or non-randomized controlled trial, with the language limited to Chinese and English; (2) Participants of the study: pediatric patients (aged <18 years) who underwent enemas due to intussusception; (3) Interventions adopted: fluoroscopic air enema or ultrasound-guided saline enemas were used; (4) The main outcome indicators: time to reset, success rate of reset, recurrence rate, and occurrence of postoperative complications. (page 5, line 2-9)

The literature retrieval and data collection are to be carried out by at least two researchers. They will independently read the titles and abstracts of the literature, excluding those that are not relevant to the content of this study. Subsequently, they will carefully read the full texts according to inclusion and exclusion criteria, extracting relevant information. In case of disagreements, resolution will be sought through negotiation, or a third researcher may be consulted for assistance in making a judgment. (page 6, line 21-22, page 7, line 1-6) 

Q 4: Discussion: Repeat of information already in the methods and results sections. I would like a succinct summary of your main findings.

A4: Thank you very much for your valuable suggestions, in the discussion section of the article, we focus on statistically significant indicators and discuss the reasons for such results in the context of the clinic, so it is not concise enough, and We have also made certain modifications. (see the discussion in red)

Q 5: References: The referencing style is not uniform.

A4: Please accept my sincere thanks for your help with my article regarding the revision of the references, which I have accepted and revised. (see the references)

# Reviewer 2

 Q 1: The authors review the two methods of reduction in the introduction and discussion. As noted above, the included literature in the search seems to be a sub-selection without appropriate screening of all literature identified by the initial search. I would request clarification regarding the 986 articles that were identified but expunged prior to screening. I would suggest the authors screen these 986 articles using their inclusion and exclusion criteria to ensure all relevant literature is considered.

A1: Please accept my sincere thanks for your help with my article, we developed inclusion and exclusion criteria based on the PICOS principles and conducted a comprehensive literature search in which two investigators independently read article titles and abstracts to initially exclude literature that did not match the content of this study, with specific reasons for exclusion demonstrated (see the suppl-figure 1 and revised-Figure 1 in the manuscript), and we have revised the text in red (page 5, line 2-9).

Q 2: It is helpful that the authors have assessed the risk of bias for individual studies. I would consider highlighting in the discussion that the literature is based on a mix of randomized and non-randomized trials. Specifically, the fact that one very large retrospective cohort study contributes 1119 patients to the total of 9391 (and most of the included randomized trials are much smaller than the non-randomized). Although this information is available in Table 1, I would suggest making it more accessible to readers.

A2: I am so grateful for the valuable advice, we have made changes in the discussion section and added that element. (page 15, line 14-15)

Q 3: Overall this is a well-written manuscript and the comparison of UGHR and FGAR is worthwhile; however, revision is required to include screening of all appropriate literature as there may be selection bias with the current methods that do not explain why only a subset of literature underwent screening.

A 3: Thank you very much for your valuable suggestions, As described in the text, our literature search strategy was to abstract or title all literature including "intussusception", "enema", "ultrasound-guided", so by reading the title and abstract of those articles, it was possible to exclude most of the literature. For example, studies in which the subject was not a child, uncontrolled studies (only UGHR or FGAR), or fluoroscopic or ultrasound-guided enemas with other fluid will be excluded after screening. (see the suppl-figure 1 and revised-Figure 1 in the manuscript)

---

## [Editor Report · Decision Letter 1]

16 Jan 2024

Air enema reduction versus hydrostatic enema reduction for intussusceptions in children: An systematic review and meta-analysis

PONE-D-23-36429R1

Dear Dr. Xie,

We’re pleased to inform you that your manuscript has been judged scientifically suitable for publication and will be formally accepted for publication once it meets all outstanding technical requirements.

Kind regards,

Ozlem Boybeyi-Turer

Academic Editor

PLOS ONE
---

## [Editor Report · Acceptance letter]

8 Mar 2024

PONE-D-23-36429R1 

PLOS ONE

Dear Dr. Xie, 

I'm pleased to inform you that your manuscript has been deemed suitable for publication in PLOS ONE. Congratulations! Your manuscript is now being handed over to our production team.

Kind regards, 

on behalf of

Professor Ozlem Boybeyi-Turer 

Academic Editor

PLOS ONE